# NASH EQUILIBRIA IN REWARD-POTENTIAL MARKOV GAMES:
# ALGORITHMS, COMPLEXITY, AND APPLICATIONS

## ABSTRACT

Markov games that exhibit potential functions for rewards in each state, referred to as Reward-Potential Markov Games (RPMGs), do not inherently qualify as Markov Potential Games (MPGs), which require state-dependent potential functions for value functions. This discrepancy, widely acknowledged in recent literature on MPGs, remains highly unexplored. RPMGs, with their easier-to-verify and arguably more minimal reward-potential property, have not received adequate attention. We embark on the exploration of RPMGs, observing that computing a stationary Nash equilibrium (NE) is PPAD-hard for infinite-horizon RPMGs, even under constraints on transition functions. In contrast to results on stationary equilibria in Markov games, we establish that computing a nonstationary Nash equilibrium in finite-horizon RPMGs is PPAD-hard without any assumptions on transition functions. On a positive note, we present an algorithm capable of breaking curse of multiagents by efficiently computing an $\epsilon$-approximate NE in RPMGs with additive transitions, with a runtime polynomial in $1/\epsilon$. Furthermore, we extend our analysis to include an adversarial player seeking to maximize the underlying potential function, introducing the concept of Adversarial Reward-Potential Markov Games.

## 1 INTRODUCTION

The current work revolves around three main axes, *potential games*, *Markov/stochastic games*, and the *complexity of computing equilibria* in the latter games. Over time, all three subjects have claimed their fair share of attention in the literature of (algorithmic) game theory. Recent advances in the theory of Markov games have left certain questions unanswered. With the present text, we aspire to settle those questions — *i.e., what is the computational landscape of the Markovian extensions of potential games when one does not assume the existence of a potential function for the values, rather one only assumes it for the rewards of each state?* Before proceeding to answer, let us introduce some context.

*Potential games* (Monderer & Shapley, 1996; Rosenthal, 1973) have enduringly reserved a central role in the theory of games. They enjoy an array of favorable mathematical properties and are able to model an abundance of real-world applications. Roughly, they are defined as games in which deviations in the utility of any agent —when they unilaterally deviate— can be tracked by a single function, the *potential* function. Much of the research of algorithmic game theory revolves around devising algorithms that can compute equilibria in such games, and arguing about the computational complexity of that task. Before substantial progress had been made, research dealt with the framework of static and normal-form games that do not allow change in the game itself. Arguably, a good portion of the initial issues (Babichenko & Rubinstein, 2021; Fearnley et al., 2022) has been settled and now researchers are investigating games that are allowed to change over time. This is where Markov games enter the frame; they are games with an inherent dynamic nature.

*Markov games* (MGs) — or stochastic games — (Shapley, 1953) are a generalization of multi-agent Markov decision processes (MDPs). The joint action of all players affects the transitions of the process and not just the individual instantaneous rewards of each agent. MGs stand as the theoretical framework for the purpose of rigorously formulating and addressing questions in field of multi-agent

reinforcement learning (MARL) (Littman, 1994). A computational issue which has been encountered by MARL literature is the *curse of multiagents*. Effectively, the curse of multiagents signifies an algorithmic complexity of achieving a given objective (*e.g.* computing an equilibrium) that depends exponentially on the number of agents and/or each agent's actions.

*The complexity of computing a Nash equilibrium* is a central topic in algorithmic game theory. Far from being a peculiar intellectual pursuit, advances such as proving the intractability of computing Nash equilibria in general games have challenged the then-established credo of economists that markets reach and operate in equilibrium states (Papadimitriou, 2014). The complexity class PPAD (Papadimitriou, 1994) characterizes problems that belong in the class NP but whose solutions are guaranteed to exist due to a fixed point argument — particularly, using the Brouwer fixed point theorem. Approximating Nash equilibria in two-player general-sum games is known to be PPAD-complete, (Daskalakis et al., 2009; Chen et al., 2009) and it is highly unlikely that an $\epsilon$-approximate equilibrium can be approximate in time that is polynomial in $1/\epsilon$. Nevertheless, not all games assume full generality and, as such, the equilibrium intractability results do not apply to them *Structure* in the game does not reduce it into triviality and still poses theoretical challenges when designing algorithms to approximate equilibria. Apart from potential games, research has focused on *two-player zero-sum games*, *(strategically) zero-sum polymatrix games*, *adversarial team games*, and *monotone games*; for all the latter, there have been contributed algorithms and learning dynamics that approximate equilibria with a varying degree of efficiency that is mostly favorable.

**Outline of our contributions.**   In the current work, we address the underlying question posed in a number of recent papers that concern the Markovian or stochastic extension of static potential games, (Leonardos et al., 2021; Lin et al., 2020; Mguni et al., 2021),

$$\text{Is the assumption of rewards that exhibit a potential function} \atop \text{enough for the tractability of equilibria in Markov games?} \qquad (\star)$$

In a nutshell, we prove that when no assumption holds for the transitions, even *nonstationary* approximate equilibria are PPAD-hard to compute — regardless of the reward-potential assumption (Theorem 3.2) and in contrast to recent results that concern (Markovian) stationary approximate equilibria of infinite-horizon games (Daskalakis et al., 2022; Jin et al., 2022; Deng et al., 2023). We observe how the latter results can be utilized to derive the PPAD-hardness of approximate equilibria in reward-potential games even when the transition functions are restricted from attaining full generality (Observation 1).

After concluding that a certain assumption on the transitions is necessary, we consider reward-potential MGs with *additive transitions*, the most general assumption we are allowed to hold in light of our hardness results. We manage to design an efficient algorithm for computing NE that runs in time polynomial in $1/\epsilon$ and $H$, where $H$ is the horizon of the game. We then extend our results to the class of *adversarial reward-potential* MGs (Theorem 3.3).

## 2  PRELIMINARIES

In this section, we will introduce the framework of Markov games (MGs), and furthermore restate some preliminary definitions of potential and other kinds of games relevant to our work.

**Notation.**   We will denote $[n] := \{1, \ldots, n\}$. A boldface is used for matrices and vectors, while scalars are denoted using a lightface font. Unless stated otherwise $\|\cdot\| := \|\cdot\|_2$. The $O(\cdot)$ might be used to suppress polynomial dependencies on the natural parameters of the game. $\Delta(\mathcal{A})$ denotes the simplex of support $\mathcal{A}$.

### 2.1  NORMAL-FORM POTENTIAL GAMES

As a reminder, we define normal-form potential games. A normal-form game is the tuple $\mathcal{G}\left(n, \{\mathcal{A}_i\}_{i\in[n]} \{u_i\}_{i\in[n]}\right)$; every player $i$ is endowed with pure strategies $a_i \in \mathcal{A}_i$; their mixed strategies are denoted as $\boldsymbol{x}_i \in \Delta(\mathcal{A}_i)$, and we mark $\boldsymbol{x} := (\boldsymbol{x}_1, \ldots, \boldsymbol{x}_n)$. The utility of player $i$ is denoted as $u_i(\boldsymbol{x})$. A potential game is a game that asserts a function $\psi : \prod_{i=1}^{n} \mathcal{A}_i \to \mathbb{R}$, such that $\forall \boldsymbol{x} \in \prod_{i=1}^{n} \Delta(\mathcal{A}_i), \forall i \in [n], \forall \boldsymbol{x}_i' \in \Delta(\mathcal{A}_i)$

$$\psi(\boldsymbol{x}_i', \boldsymbol{x}_{-i}) - \psi(\boldsymbol{x}) = u_i(\boldsymbol{x}_i', \boldsymbol{x}_{-i}) - u_i(\boldsymbol{x}).$$

## 2.2 MARKOV GAMES

Following, we present the framework of MGs in both finite and infinite-horizon and then proceed to define value functions and equilibrium notions. First, we note that an $n$-player MG consists of a tuple $\Gamma\left(n, H, \mathcal{S}, \{\mathcal{A}_i\}_{i\in[n]}, \mathbb{P}, \{r_i\}_{i\in[n]}, \gamma, \boldsymbol{\rho}\right)$. In particular,

- $H \in \mathbb{N}_+$ stands for the *time horizon*, or the length of every episode of the game,

- $\mathcal{S}$, is the finite state space whose cardinality is denoted as $S := |\mathcal{S}|$

- $\{\mathcal{A}_i\}_{i\in[n]}$ is the set of action spaces of the players, and $\mathcal{A} := \mathcal{A}_1 \times \cdots \times \mathcal{A}_n$ stands for the *joint action space*; moreover, a *joint action* will generally be noted as $\boldsymbol{a} = (a_1, \ldots, a_n) \in \mathcal{A}$,

- $\mathbb{P} := \{\mathbb{P}_h\}_{h\in[H]}$ is the set of all *transition kernels*, with $\mathbb{P}_h : \mathcal{S} \times \mathcal{A} \to \Delta(\mathcal{S})$; further, $\mathbb{P}_h(\cdot|s, \boldsymbol{a})$ denotes the probability of transitioning to a state of the state space conditioned on the joint action $\boldsymbol{a}$ being selected at time $h$ and state $s$ — in infinite-horizon games $\mathbb{P}$ does not depend on $h$ and we drop the index,

- $r_i := \{r_{i,h}\}$ is the reward function of player $i$ at time $h$; $r_{i,h} : \mathcal{S}, \mathcal{A} \to [-1, 1]$ yields the reward of player $i$ at a given state and joint action — in infinite-horizon games, $r_{i,h}$ is the same for every $h$ and the subscript is dropped,

- a discount factor $\gamma \in [0, 1]$, which is generally set to 1 when $H < \infty$, and $\gamma < 1$ when $H \to \infty$,

- an initial state distribution $\boldsymbol{\rho} \in \Delta(\mathcal{S})$.

Given a MG, $\Gamma$, we define the $(s, h)$-*subgame*, $\Gamma_{s,h}$, as the game that inherits every element of game $\Gamma$—reward functions, transitions, *etc.*— starting at time step $h \in [H]$ and state $s \in \mathcal{S}$.

## 2.3 POLICIES, VALUE FUNCTIONS, AND EQUILIBRIA

We commence with the remark that all the different notions of equlibria to be defined are guaranteed to always exist (Fink, 1964; Solan & Vieille, 2015). Before proceeding to the definition of different notions of equilibria in MGs, one needs to present the different kinds of individual *policies* followed by players. There are two main dichotomies of policies in the contemporary literature of MARL. A policy can be *stationary* or *nonstationary*, and *Markovian* or *non-Markovian*. Nonstationary policies of player $i$, $\boldsymbol{\pi}_i$, are allowed to change depending on the time step of the horizon of the game, while stationary policies, $\boldsymbol{\pi}_i$ attribute the same probability distribution over actions in every state of the game. Policies that are allowed to take into account past information of the game are known as non-Markovian, while policies that depend only on the state and the time step of the horizon are known as Markovian.

Moreover, policies that attribute a distribution of *joint* action in every state, *i.e., joint policies* can be *correlated* or *product* policies. A joint policy $\boldsymbol{\pi}$ is said to be a product one when there exist individual policies $\{\boldsymbol{\pi}_i\}_{i\in[n]}$ such that $\boldsymbol{\pi} = \boldsymbol{\pi}_1 \times \cdots \times \boldsymbol{\pi}_n$. Of course, product policies are a strict subset of correlated policies. A policy that assigns probability 1 to a single action in every state $s$ (and timestep $h$ if it is nonstationary) is called *deterministic*.

### 2.3.1 THE FINITE HORIZON

In the finite horizon setting, the game lasts for a finite amount of steps, $H < \infty$. Typically, in this setting, policies are defined to be *nonstationary* as even in a single-agent finite-horizon MDP, the optimal stationary policy can be arbitrarily worse than an optimal nonstationary policy.

**Policies.** In detail, a (nonstationary Markovian) policy of player $i$, $\boldsymbol{\pi}_i := \{\boldsymbol{\pi}_{i,s,h} \in \Delta(\mathcal{A}_i)\} \in \Delta(\mathcal{A}_i)^{\mathcal{S}\times[H]}$ attributes a probability of playing an action $a \in \mathcal{A}_i$ at timestep $h \in [H]$ and state $s \in \mathcal{S}$. Further, we denote a joint policy by dropping the subscript, *i.e.,* $\boldsymbol{\pi} := \{\boldsymbol{\pi}_{s,h} \in \Delta(\mathcal{A})\} \in \Delta(\mathcal{A})^{\mathcal{S}\times[H]}$. A joint policy is possibly correlated as it is allowed to belong to the simplex of joint actions for every $s$ and $h$. We overload notation to note $r_{i,h}(s, \boldsymbol{\pi}) = \mathbb{E}_{\boldsymbol{a}\sim\boldsymbol{\pi}}[r_{i,h}(s, \boldsymbol{a})]$ and $\mathbb{P}_h(s, \boldsymbol{\pi}) = \mathbb{E}_{\boldsymbol{a}\sim\boldsymbol{\pi}}[r_{i,h}(s, \boldsymbol{a})]$ accordingly.

**Value function.** Typically, the discount factor $\gamma$ is set to 1 in finite-horizon MGs. As such, we define the value function of player $i$ in a finite horizon MG to be the expected cumulative reward when the game starts at state $s_1$ and time step $h = 1$,

$$V_{i,h}^{\boldsymbol{\pi}}(s_1) := \mathbb{E}_{\boldsymbol{\pi}} \left[ \sum_{\tau=h}^{H} r_{i,\tau}(s_\tau, \boldsymbol{a}_\tau) \,\Big|\, s_1 \right].$$

**Notions of equilibria.** A best-response policy of player $i$ to $\boldsymbol{\pi}_{-i}$ will be noted as $\boldsymbol{\pi}_i^\dagger \in \arg\max_{\boldsymbol{\pi}_i'} V_{i,1}^{\boldsymbol{\pi}_i' \times \boldsymbol{\pi}_{-i}}(s_1)$, $\boldsymbol{\pi}_i' \in \Delta(\mathcal{A}_i)^{\mathcal{S} \times [H]}$. Moreover, the value of the best-responding policy of player $i$ is noted as $V_{i,1}^{\dagger, \boldsymbol{\pi}_{-i}}(s_1) := \max_{\boldsymbol{\pi}_i'} V_{i,1}^{\boldsymbol{\pi}_i' \times \boldsymbol{\pi}_{-i}}(s_1)$, where $\boldsymbol{\pi}_i' \in \Delta(\mathcal{A}_i)^{\mathcal{S} \times [H]}$. We will only define the nonstationary Markovian NE.

**Definition 2.1** (NE—nonstationary)**.** *For an $\epsilon \geq 0$, a joint product policy $\boldsymbol{\pi} \in \prod_{i=1}^{n} \Delta(\mathcal{A}_i)^{\mathcal{S} \times [H]}$ is*

- *an $\epsilon$-approximate Markov-perfect coarse Nash equilibrium if,*

$$V_{i,h}^{\dagger, \boldsymbol{\pi}_{-i}}(s) - V_{i,h}^{\boldsymbol{\pi}}(s) \leq \epsilon, \; \forall i \in [n], s \in \mathcal{S}, h \in [H],$$

- *an $\epsilon$-approximate (Markov) coarse correlated equilibrium if,*

$$V_{i,1}^{\dagger, \boldsymbol{\pi}_{-i}}(s_1) - V_{i,1}^{\boldsymbol{\pi}}(s_1) \leq \epsilon, \; \forall i \in [n].$$

### 2.3.2 THE INFINITE HORIZON

When the horizon of an MG is infinite, *i.e.*, $H \to \infty$, the policies that are sought after are typically stationary. Reward and transition functions do not depend on time and as such the subscript $h$ is dropped, $r_h = r$, $\mathbb{P}_h = \mathbb{P}_h, \forall h \in [H]$. There are two standard ways of defining value functions in infinite-horizon games, *undiscounted average reward* and *discounted cumulative reward*. The latter predominates contemporary literature of infinite-horizon MGs and it is the one we will define here.

**Policies.** For player $i$, Markovian stationary policy $\boldsymbol{\pi}_i \in \Delta(\mathcal{A}_i)^{\mathcal{S}}$ attributes a distribution over actions in every state regardless of the time step of the horizon. Similarly, a stationary joint policy is defined as $\boldsymbol{\pi} \in \Delta(\mathcal{A})^{\mathcal{S}}$.

**Value functions.** Given a joint policy $\boldsymbol{\pi}$, the value function of player $i$ is defined as the average discounted cumulative reward,

$$V_i^{\boldsymbol{\pi}}(s_1) := \mathbb{E}_{\boldsymbol{\pi}} \left[ \sum_{\tau=1}^{\infty} \gamma^{\tau-1} r_i(s_\tau, \boldsymbol{a}_\tau) \,\Big|\, s_1 \right].$$

Moreover, slightly abusing notation we denote $V_i^{\boldsymbol{\pi}}(\boldsymbol{\rho}) = \mathbb{E}_{s_1 \sim \boldsymbol{\rho}} \left[ V_i^{\boldsymbol{\pi}}(s_1) \right]$.

Additionally, a best-response policy of player $i$ to the potentially correlated policy $\boldsymbol{\pi}_{-i}$ is denoted as $\boldsymbol{\pi}_i^\dagger \in \arg\max_{\boldsymbol{\pi}_i'} V_i(\boldsymbol{\rho}), \boldsymbol{\pi}_i' \in \Delta(\mathcal{A}_i)^{\mathcal{S}}$. Finally, the value of the best-responding policy of player $i$ is noted as $V_i^{\dagger, \boldsymbol{\pi}_{-i}}(\boldsymbol{\rho}) = \max_{\boldsymbol{\pi}'} V_i^{\boldsymbol{\pi}' \times \boldsymbol{\pi}_{-i}}(\boldsymbol{\rho}), \boldsymbol{\pi}' \in \Delta(\mathcal{A}_i)^{\mathcal{S}}$.

**Notions of equilibria.** Analogous to the finite-horizon MGs, infinite-horizon MGs assert an array of equilibria that are guaranteed to exist. We will define the notions that are relevant, namely approxiamte CCEs and approximate NEs.

**Definition 2.2** (CCE—stationary)**.** *For an $\epsilon \geq 0$, a joint product policy $\boldsymbol{\pi} \in \Delta(\mathcal{A})^{\mathcal{S}}$ is*

- *an $\epsilon$-approximate Markov-perfect coarse coarse correlated equilibrium if,*

$$V_i^{\dagger, \boldsymbol{\pi}_{-i}}(s) - V_i^{\boldsymbol{\pi}}(s) \leq \epsilon, \; \forall i \in [n],$$

- *an $\epsilon$-approximate (Markov) coarse correlated equilibrium if,*

$$V_i^{\dagger, \boldsymbol{\pi}_{-i}}(\boldsymbol{\rho}) - V_i^{\boldsymbol{\pi}}(\boldsymbol{\rho}) \leq \epsilon, \; \forall i \in [n].$$

**Definition 2.3** (NE—stationary). *For an $\epsilon \geq 0$, a joint product policy $\boldsymbol{\pi} \in \prod_{i=1}^{n} \Delta(\mathcal{A}_i)^{\mathcal{S}}$ is*

- *an $\epsilon$-approximate Markov-perfect coarse Nash equilibrium if,*

$$V_i^{\dagger, \boldsymbol{\pi}_{-i}}(s) - V_i^{\boldsymbol{\pi}}(s) \leq \epsilon, \ \forall i \in [n],$$

- *an $\epsilon$-approximate (Markov) Nash equilibrium if,*

$$V_i^{\dagger, \boldsymbol{\pi}_{-i}}(\boldsymbol{\rho}) - V_i^{\boldsymbol{\pi}}(\boldsymbol{\rho}) \leq \epsilon, \ \forall i \in [n].$$

## 2.4 Markov Games with Structure

Here, we will provide a short exposition on different structures applied to the primitives of the MG, *i.e.,* the reward and transition functions.

**Warm-up: Markov potential games.** An important class of MGs that has gained traction in recent literature is the class of Markov potential games (MPGs) (Leonardos et al., 2021; Zhang et al., 2021; Mguni et al., 2021). In this class of games, there exists a state-dependent potential function for the *value functions* of the players, rather than just the reward functions. In (Leonardos et al., 2021) it is highlighted that an MPG can be zero-sum in the rewards of one state and potential in the rewards of another. We remark that a MPG, it is assumed that there exists a potential function for the *value functions* of the game, rather than the rewards. One is encouraged to revise the counterexamples provded in (Leonardos et al., 2021; Zhang et al., 2021) for MGs which fail to be an MPG even though every stage game is a potential game, or MGs with stage games which are zero-sum games, yet they are MPGs.

**Definition 2.4** (Markov potential game — MPG). *An MG is a Markov potential game if there exists a state-dependent potential function, $\Phi^{\boldsymbol{\pi}}(s)$, such that for all players $i \in [n]$, joint policies $\boldsymbol{\pi}$, and unilateral deviations $\boldsymbol{\pi}_i'$,*

$$\Phi^{\boldsymbol{\pi}}(s) - \Phi^{\boldsymbol{\pi}_i', \boldsymbol{\pi}_{-i}}(s) = V_i^{\boldsymbol{\pi}}(s) - V_i^{\boldsymbol{\pi}_i', \boldsymbol{\pi}_{-i}}(s).$$

### 2.4.1 Structured Rewards

**Reward-potential Markov games.** The class of reward-potential MGs is defined to be the MGs whose rewards in every state are characterized by the existence of a potential function. *I.e.,* given a joint policy, changes in the utility of each player, when they unilaterally deviate, are described by the differences in the potential function.

**Remark 1.** *In our opinion, this is a justified and reasonable alternative Markovian extension of the class of potential games. Further, the proposed assumption is rather minimal, a lot more so than the existence of a potential function for the value functions of the players.*

**Definition 2.5** (Reward-potential Markov game — RPMG). *We call a Markov game reward-potential when for every state $s$ (and timestep $h$ of the horizon), there exists a function $\phi_h : \mathcal{S} \times \Delta(\mathcal{A}) \to \mathbb{R}$ such that for all players $i \in [n]$, joint policies $\boldsymbol{\pi} \in \Delta(\mathcal{A})$, and unilateral deviations $\boldsymbol{\pi}_i' \in \Delta(\mathcal{A}_i)$,*

$$\phi_h(s, \boldsymbol{\pi}) - \phi_h(s, \boldsymbol{\pi}_i', \boldsymbol{\pi}_{-i}) = r_{i,h}(s, \boldsymbol{\pi}) - r_{i,h}(s, \boldsymbol{\pi}_i', \boldsymbol{\pi}_{-i}).$$

**Adversarial reward-potential Markov games.** Inspired by the setting proposed in (Babaioff et al., 2007) and more recently studied by Anagnostides et al. (2023); Orzech & Rinard (2023), it is possible to further extend RPMGs to MGs whose rewards follow an *adversarial potential* structure. This means that the $n + 1$ players of the game are split into a group of $n$ agents and an *adversarial player*; the reward functions of the group of the first $n$ players are characterized by a potential function — given that the strategy of the adversary remains fixed. The adversarial player's reward function is precisely the opposite value of the group's potential function. Particularly, we define the following class of games:

**Definition 2.6** (Adversarial reward-potential Markov game — ARPMG). *An adversarial reward-potential Markov game is an MG with $n + 1$ players. There exists a function $\phi_h : \mathcal{S} \times \Delta(\mathcal{A}) \to \mathbb{R}$ such that for all players of the group, $i \in [n]$, joint policies $\boldsymbol{\pi} \in \Delta(\mathcal{A})$, and unilateral deviations $\boldsymbol{\pi}_i' \in \Delta(\mathcal{A}_i)$,*

$$\phi_h(s, \boldsymbol{\pi}) - \phi_h(s, \boldsymbol{\pi}_i', \boldsymbol{\pi}_{-i}) = r_{i,h}(s, \boldsymbol{\pi}) - r_{i,h}(s, \boldsymbol{\pi}_i', \boldsymbol{\pi}_{-i}).$$

*Additionally, the reward function of the adversary is defined as:*

$$r_{\text{adv},h}(s,\boldsymbol{\pi}) = \phi_h(s,\boldsymbol{\pi}).$$

**Remark 2.** *We note that this class of MGs differs from the adversarial potential Markov games defined in (Kalogiannis et al., 2022); the latter setting assumes the existence of a potential function for the value functions of the players of the team rather than just their reward functions.*

### 2.4.2 STRUCTURED TRANSITIONS

As we intend to demonstrate, the transition function can essentially be used to simulate any general-sum normal form game even when the reward function form a potential game. This goes to show that computing approximate stationary equilibria is not only hard in infinite-horizon games; transition functions in their full generality can make even finite-horizon nonstationary equilibria intractable. As such, we will present several assumptions that are standard in the literature of MGs and we shall see that under those, approximating equilibria is a tractable problem. We will highlight the structural assumptions of (i) *a single controller*, (ii) *switching-control*, and (iii) *additive transitions*. Each of these assumptions is strictly contained to the one that follows it.

$$\text{single controller} \subset \text{switching control} \subset \text{additive transitions}.$$

**Single controller.** The single controller assumption in words translates to the fact that only one player out of the many of a MG can affect the transitions from one state to another. This assumption is one that has been studied extensively in past as well as contemporary literature (Parthasarathy & Raghavan, 1981; Sayin et al., 2022).

**Switching control.** A more slightly more general assumption on the structure of the transitions is that of switching control (Vrieze et al., 1983; Mohan & Raghavan, 1987; Kalogiannis & Panageas, 2023). When an $n$-player MG is characterized by switching control, the state-space is divided into disjoint subsets $\{\mathcal{S}_i\}_{i\in[n]}$, with $\mathcal{S} = \cup_{i=1}^n \mathcal{S}_i$; in every such set $\mathcal{S}_i$, it is only player $i$ that controls the transitions.

**Additive transitions.** Finally, the more general transition structure we will present is that of additive transitions. This structure contains all previous assumptions as special cases and has been investigated in an array of works (Raghavan et al., 1985; Flesch et al., 2007; Park et al., 2023). It can be seen as inducing an interpolation between independent (or, *product*) state-space games (Flesch et al., 2008) and standard MGs.

**Definition 2.7** (Additive transitions). *A Markov game is said to exhibit additive transitions when in every state $s$ and timestep $h$ of the horizon, it holds that,*

$$\mathbb{P}_h(s'|s,\boldsymbol{a}) = \sum_{i\in[n]} \omega_{i,s,h}\,\mathbb{P}_{i,h}(s'|s,a_i),$$

*where $\omega_{i,s,h} \geq 0, \forall i \in [n]$ and $\sum_{i\in[n]} \omega_{i,s,h} = 1$.*[1]

### 2.4.3 AN EXAMPLE

**Turn-based MGs.** *Turn-based* MGs are a class of structured MGs that has proven useful in advancing the understanding of the computational complexity of equlibria in MGs (Daskalakis et al., 2022; Jin et al., 2022; Deng et al., 2023).

**Definition 2.8** (Turn-based Markov game—TBMG). *In an $n$-player turn-based MG, the state space $\mathcal{S}$ is split into disjoint sets $\{\mathcal{S}_i\}_{i\in[n]}$. In every such set $\mathcal{S}_i$, player $i$ (called the controller) determines entirely through their actions both the transitions and the reward functions of all players.*

One can observe that turn-based MGs are a special case of MGs with switching control. Further, correlated policies are equivalent to product policies in those games, making CCEs and NEs equivalent may they be stationary or nonstationary and perfect or not. We will refer to them as equilibria without further specification.

---

[1]When, $\omega_{s,h,j} = 1$ and $\omega_{s,h,i} = 0, \forall k \neq i$ we retrieve the switching-control setting.

## 3 MAIN RESULTS

In this section, we demonstrate the necessity of assuming *additive transitions* in RPMGs even for computing *nonstationary* approximate equilibria. Then, we present Algorithm 1 which computes NE in RPMGs with additive transitions. We highlight that our results concern *nonstationary* equilibria and not only stationary ones.

### 3.1 HARDNESS RESULTS

We commence this subsection by citing a recent result regarding the computational complexity of computing *stationary* equilibria in infinite-horizon MGs which of course has implications for RPMGs.

**Theorem 3.1** (PPAD-hardness for perfect equilibria — (Daskalakis et al., 2022; Jin et al., 2022; Deng et al., 2023)). *There exists a constant $\epsilon > 0$ such that the problem of computing an $\epsilon$-approximate perfect NE in 2-player, turn-based stochastic games wiht $\gamma = 1/2$ is PPAD-hard. As such, the problem of computing an $\epsilon$-approximate perfect CCE in 2-player, infinite-horizon stochastic games with $\gamma = 1/2$ is PPAD-hard.*

**Observation 1.** *Computing an $\epsilon$-approximate stationary CCE in reward-potential Markov games is PPAD-hard.*

Let us make the latter observation clearer. We denote the controller of state $s \in \mathcal{S}_i$, $\mathrm{cr}(s) = i$. From the definition of TBMG, there exist functions $r'_j$ for each player $j$, such that $r_j(s, \boldsymbol{a}) = r'_j(s, a_{\mathrm{cr}(s)})$. Similarly, there exist $\mathbb{P}'$ such that $\mathbb{P}(s'|s, \boldsymbol{a}) = \mathbb{P}'(s'|s, a_{\mathrm{cr}(s)})$.

Now, we can observe that in a TBMG, the sum of rewards in every state is trivially a potential function for the rewards of that state,

$$\phi(s, \boldsymbol{a}) = \sum_{i \in [n]} r_i(s, \boldsymbol{a}) = \sum_{i \in [n]} r'_i\left(s, a_{\mathrm{cr}(s)}\right).$$

*i.e.,* it holds that,

$$\phi(s, a'_j, \boldsymbol{a}_{-j}) - \phi(s, \boldsymbol{a}) = r(s, a'_j, \boldsymbol{a}_{-j}) - r(s, \boldsymbol{a}).$$

Hence, TBMGs are in fact a special case of reward-potential Markov games. Next, we show that when transitions assert full generality, even the computation of *nonstationary* approximate NE is PPAD-hard for *finite-horizon* games. Our main complexity contribution is that:

**Theorem 3.2.** *Computing a nonstationary Markovian $\epsilon$-approximate NE policy in reward-potential Markov games is PPAD-hard.*

**Proof.** Consider a 2-player general-sum game $\Gamma$ with payoff matrices $(\mathbf{U}, \mathbf{V})$ for player 1, 2 accordingly. Pure strategies of players 1 and 2 are denoted $a_i, b_j$, accordingly, with $i \in [m]$ and $j \in [n]$. Hence, $\mathbf{U}, \mathbf{V} \in \mathbb{R}^{m \times n}$.

We construct a 2-player reward-potential Markov game $\Gamma'$ as follows:

- the time horizon of the game is $H = 3$,

- players 1, 2 have the same set of available actions as players in game $\Gamma$; $\{a_i\}_{i \in [m]}, \{b_j\}_{j \in [n]}$,

- there is an initial state $s_0$,

- for every pair of actions $a_i, b_j$ of the initial game there is a state $s_{ij}$; *i.e.,* $\mathcal{S} = \{s_{ij}, \ ij \in [m] \times [n]\}$

- in state $s_{ij}$ player 1 gets reward $U_{ij}$, player 2 gets $V_{ij}$; in $s_0$, they both get reward 0,

- transitions are deterministic and $\mathbb{P}(s_{ij}|s_0, a_i, b_j) = 1$, while states $s_{ij}$ are absorbing.

The value functions of players 1, 2 for policies in $s_0$ $\boldsymbol{x} := \boldsymbol{\pi}_1(s_0, h = 1), \boldsymbol{y} =: \boldsymbol{\pi}_2(s_0, h = 1)$ are:

$$\begin{cases} V_1(s_0) &= 0 + \sum_{a,b} \sum_{s_{ij} \in \mathcal{S}} x(a)y(b) \, \mathbb{P}(s_{ij}|s_0, a, b) U_{ij} \\ &= \sum x(a_j)y(b_j)U_{ij} = \boldsymbol{x}^\top \mathbf{U} \boldsymbol{y} \\ V_2(s_0) &= \boldsymbol{x}^\top \mathbf{V} \boldsymbol{y}. \end{cases}$$

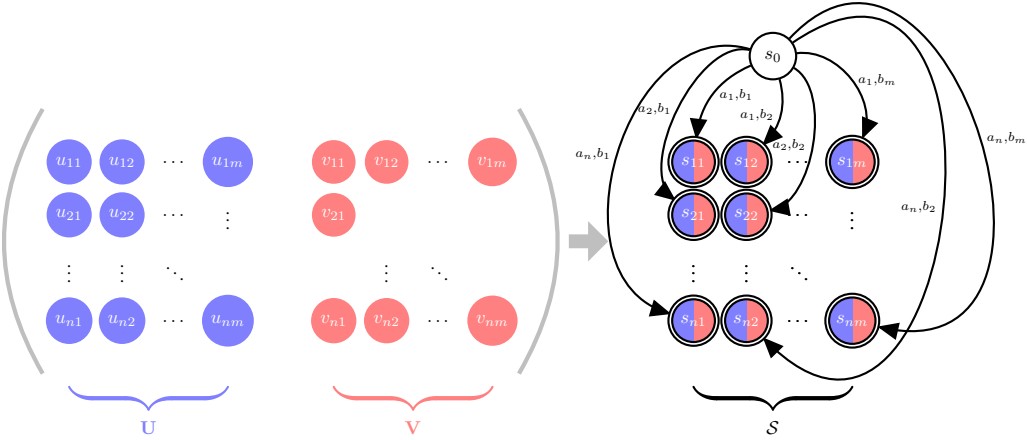

Figure 1: Illustration of the construction used for the PPAD-hardness of **nonstationary** NE.

Hence, Nash equilibria of game $\Gamma$ coincide with the $\boldsymbol{x}, \boldsymbol{y}$ policies of Nash equilibria in game $\Gamma'$ and the complexity of approximating them is known due to (Chen et al., 2009; Daskalakis et al., 2009). □

## 3.2 REWARD-POTENTIAL MARKOV GAMES

Having decisively proven the necessity of assuming a structure on the transitions of the game, we state our main algorithmic result for RPGMs with *additive transitions*.

**Theorem 3.3** (Informal version of Theorem D.3). *Algorithm 1 computes an $\epsilon$-approximate nonstationary NE for an RPMG with additive transitions in time $O(H^5 |\mathcal{S}|^2 / \epsilon^2)$.*

---

**Algorithm 1** Backwards-Inductive NE Computation in Reward-Potential MGs

---

1: **input:** $n, \mathcal{S}, H$ and accuracy parameter $\epsilon$.
2: **initialization:** $\hat{\boldsymbol{V}}_{i,H} = \boldsymbol{0}$ for all agents $i \in [n]$
3: **for** $h = H - 1$ to $1$ **do**
4:     `\\Approx. NE for subgame` $\Gamma_{s,h}$ `for all` $s$ `with accuracy` $\epsilon/H$
    $\boldsymbol{x}_{s,h} \leftarrow \mathsf{NE\text{-}Oracle}\left(\frac{\epsilon}{H}, \left\{\boldsymbol{r}_h, \mathbb{P}_h, \hat{\boldsymbol{V}}_{h+1}\right\}\right)$ `\\for all` $s \in \mathcal{S}$
5:     `\\Update value function`
    $\hat{\boldsymbol{V}}_{i,s,h} \leftarrow \boldsymbol{r}_{i,h}(s, \boldsymbol{x}_h) + \mathbb{P}_h(s, \boldsymbol{x}_h)\hat{\boldsymbol{V}}_{i,s,h+1}$
6: **end for**
7: **return** $\{\boldsymbol{x}_h\}_{h \in [H]}$

---

**Properties of RMPGs.** We conclude this subsection by noting an interesting property of RPMGs. They do inherit the property of asserting pure NEs from their counterpart in nomral and static form. In the case that it was desirable, we could modify the implementation of NE-Oracle in Algorithm 1 in such that could compute pure NE in every state and also retrieve *deterministic* nonstationary NE policies for RPMGs.

**Theorem 3.4.** *Finite-horizon reward-potential games with additive transitions assert pure Nash equilibria.*

A further note we would like to include is the fact that infinite-horizon RPMGs attain *deterministic* approximate nonstationary equilibria by the standard trick of truncating the horizon of the game. Namely, we set $H = \frac{\log(1/\epsilon)}{1-\gamma}$ and modifying the reward functions such that $r_{i,h}(s, \cdot) = \gamma^{h-1} r_i(s, \cdot)$.

**Corollary 3.1.** Infinite-horizon RPMGs with discount parameter $\gamma$, attain a deterministic nonstationary approximate NE that can be computed in time poly $\left(\frac{1}{\epsilon}, \frac{1}{1-\gamma}\right)$.

## 4 APPLICATIONS

We extend our results to a setting that is inherently tied to an underlying *potential function*, namely *adversarial reward-potential Markov games*. In (Anagnostides et al., 2023) it is proven that the maximum of the group's potential over the adversary's actions is a potential function.

### 4.1 ADVERSARIAL REWARD-POTENTIAL MARKOV GAMES

As an extension, we consider ARPMGs, *i.e.,* MGs whose rewards follow an adversarial potential game structure. It is then straightforward to derive the following corollary from Theorem 3.2,

**Corollary 4.1.** Computing a nonstationary Markovian $\epsilon$-approximate NE policy in adversarial reward-potential Markov games is PPAD.

**Proposition 4.1.** Let an ARPMG with additive transitions, $\Gamma(n+1, H, \mathcal{S}, \mathcal{A}, \mathbb{P}, r, \gamma, \boldsymbol{\rho})$, and $\hat{\boldsymbol{V}}_{i,h+1}$ be the value vector for the $\delta$-approximate NE of the subgames $\Gamma_{s,h+1}$. Let the adversarial team normal-form games $\Gamma'_s, \forall s \in \mathcal{S}$, each with $n$ players in the team and one adversary. Define the utility function of the team to be,

$$u(s, \boldsymbol{\pi}) := \phi_h(s, \boldsymbol{\pi}) + \sum_{s' \in \mathcal{S}} \sum_{j \in [n]} \omega_{j,s,h} \, \mathbb{P}_{j,h}(\boldsymbol{\pi}_j) \hat{V}_{j,h+1}(s') - \sum_{s' \in \mathcal{S}} \omega_{\text{adv},s,h} \, \mathbb{P}_{\text{adv},h}(\boldsymbol{\pi}_{\text{adv}}) \hat{V}_{\text{adv},h+1}(s').$$

An $\epsilon$-approximate NE of each subgame $\Gamma'_s$ is also an $(\epsilon + \delta)$-approximate NE of the $\Gamma_{s,h}$ subgame.

Finally, using the algorithm of (Anagnostides et al., 2023) as a subroutine, we see that:

**Theorem 4.1.** *An $\epsilon$-approximate NE of a finite-horizon ARPMG with additive transitions can be computed in time* $\mathsf{poly}(1/\epsilon, \sum_{i \in [n+1]} |\mathcal{A}_i|, |\mathcal{S}|, H)$.

## 5 CONCLUSIONS

We examined Markov games with an assumption on the structure of rewards rather than existing stronger assumptions on the structure of individual value functions. This was a setting that was implicitly defined in many contemporary texts; yet, its computational landscape remained unexplored. We settled the question of the computational complexity of computing equilibria in such games and provided necessary assumptions for their efficient computation. We also provided corresponding algorithms. In conclusion, we would like to sketch the roadmap for some fascinating future work with the following open problems.

**Open problems.**

- Can we design *decentralized*, *rational*, and *convergent* learning algorithms that converge to a NE in additive transition RPMGs?
- Is it possible to overcome intractability using different structures on the rewards, *e.g.,* monotone rewards?
- The notion of Price of Anarchy (Koutsoupias & Papadimitriou, 1999) has been studied extensively in many classes of games including potential and smooth games (e.g., see (Roughgarden, 2009)). It would be interesting to prove price of anarchy bounds for RPMGs, extending the results of prior works that exist for MPGs (Chen et al., 2022; Zhang et al., 2023).

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

## A A SHORT REMARK ON ADDITIVE TRANSITIONS

Before proceeding any further, we would like to make it clear that *additive transitions* is the most general assumption that we can place on the transition function of a tabular MG with finite action-spaces and finite state-spaces. By definition, the transition function is a multilinear function of the individual policies. By our main theorem, Theorem 3.2, we have established that in general, bilinear transition functions can emulate any two-player general-sum normal-form game; in our construction, it is even true that the rewards will be constant in each state and independent of the actions of the players. Additive transitions result in the most general multilinear function that does not lead to intractability of equilibria and consequently the most general assumption on the transitions.

## B BACKGROUND ON MDPS AND MGS

Since MGs are a generalization of MDPs, we offer an elementary exposition of basic notions shared by both settings. We will define the value function and the action-value function (or, $Q$-function) as they play a crucial role in the theory of MDPs and MGs. Essentially, we use the framework of MGs to discuss MDPs; one just needs to consider that apart from a single agent, all other agents are dummy, *i.e.,* their actions have no effect in rewards or transitions whatsoever. We consider a MG, $\Gamma\left(n, H, \mathcal{S}, \mathcal{A}, \mathbb{P}, r, \gamma, \boldsymbol{\rho}\right)$, and define the following.

**Policies.** As previously discussed, policies can be either *stationary* or *nonstationary* and *Markovian* or *non-Markovian*. We deem only Markovian policies to be relevant in the present work and, as such, we only consider and define Markovian policies. A *stationary* policy, $\boldsymbol{\pi}_i \in \Delta(\mathcal{A}_i)^{|\mathcal{S}|}$ of agent $i \in [n]$ assigns the same distribution over actions $\mathcal{A}_i$ in every state $s \in \mathcal{S}$. On the contrary, *nonstationary* policies, $\boldsymbol{\pi}_i \in \Delta(\mathcal{A}_i)^{H \times |\mathcal{S}|}$, assign a potentially different probability distribution over states depending on the timestep of the horizon $h \in \{1, \ldots, H\}$.

**Value functions.** Given a joint policy $\boldsymbol{\pi}$, the value function of agent $i$ in a MG satisfies the *Bellman conditions*:

$$V_{i,h}^{\boldsymbol{\pi}}(s) = r_{i,h}(s, \boldsymbol{\pi}) + \sum_{s' \in \mathcal{S}} \mathbb{P}_h(s'|s, \boldsymbol{\pi}) V_{i,h+1}(s'), \ \forall h \in [H-1], s \in \mathcal{S},$$

$$V_{i,H}^{\boldsymbol{\pi}}(s) = 0, \ \forall s \in \mathcal{S}.$$

**Action-value functions.** We define the action-value function (or, q-functions), to be:

$$Q_{i,h}^{\boldsymbol{\pi}}(s, a) = r_{i,h}(s, a, \boldsymbol{\pi}_{-i}) + \sum_{s' \in \mathcal{S}} \mathbb{P}_h(s, a, \boldsymbol{\pi}_{-i}) V_{i,h+1}(s').$$

**Bellman optimality conditions.** An optimal policy $\boldsymbol{\pi}_i^\star \in \Delta(\mathcal{A}_i)^{[H] \times |\mathcal{S}|}$ satisfies the following optimality conditions,

$$V_{i,h}^{\boldsymbol{\pi}_i^\dagger, \boldsymbol{\pi}_{-i}}(s) = \max_{\boldsymbol{\pi}_i'} \left\{ r_{i,h}(s, \boldsymbol{\pi}_i', \boldsymbol{\pi}_{-i}) + \sum_{s' \in \mathcal{S}} \mathbb{P}_h(s'|s, \boldsymbol{\pi}_i', \boldsymbol{\pi}_{-i}) V_{i,h+1}(s') \right\}, \ \forall h \in [H], s \in \mathcal{S}.$$

When $\boldsymbol{\pi}_i^\dagger$ is optimal for agent $i$, then,

$$V_{i,h}^{\boldsymbol{\pi}_i^\dagger, \boldsymbol{\pi}_{-i}}(s) = \max_{a \in \mathcal{A}_i} Q_{i,h}^{\boldsymbol{\pi}_i^\dagger, \boldsymbol{\pi}_{-i}}(s, a), \ \forall h \in [H], \forall s \in \mathcal{S}.$$

**Boundedness of value.**

**Fact B.1.** Let the reward functions be bounded in $[0, 1]$, *i.e.,* $0 \leq r_h(s, \boldsymbol{a}) \leq 1$, $\forall s \in \mathcal{S}, \forall \boldsymbol{a}\mathcal{A}$, it holds that,

- $V_{i,h}(s) \leq H - h, \ \forall i \in [n], \forall h \in [H]$;

- $Q_{i,h}(s, a) \leq h, \ \forall i \in [n], \forall H - h \in [H], \forall a \in \mathcal{A}_i.$

**Lipschitz continuity of rewards and transitions.**

**Claim B.1.** In a MG $\Gamma\left(n, H, \mathcal{S}, \mathcal{A}, \mathbb{P}, \{r_i\}_{i\in[n]}, \gamma, \boldsymbol{\rho}\right)$ with additive transitions, the following inequalities hold true for any $\boldsymbol{\pi}_{s,h}, \boldsymbol{\pi}'_{s,h}$ and any $s \in \mathcal{S}$:

- $r_{i,h}(s, \boldsymbol{\pi}_{s,h}) - r_{i,h}(s, \boldsymbol{\pi}'_{s,h}) \leq \sqrt{\sum_{i\in[n]} |\mathcal{A}_i|} \left\|\boldsymbol{\pi}_{s,h} - \boldsymbol{\pi}'_{s,h}\right\|$;

- $\left|\sum_{s'\in\mathcal{S}} \left(\mathbb{P}_h(s'|s, \boldsymbol{\pi}_h) - \mathbb{P}_h(s'|s, \boldsymbol{\pi}'_h)\right) V_{i,h+1}(s')\right| \leq H|\mathcal{S}| \max_{i\in[n]} \sqrt{|\mathcal{A}_i|}$.

**Proof.** We use standard inequalities:

- Fixing any $i, s, h \in [n] \times \mathcal{S} \times [H]$, we have

$$r_{i,h}(s, \boldsymbol{\pi}) = \mathbb{E}_{\boldsymbol{a}\sim\boldsymbol{\pi}}[r_{i,h}(s, \boldsymbol{a})] = \sum_{(a_1,\ldots,a_n)\in\mathcal{A}} r_{i,h}(s, \boldsymbol{a}) \prod_{i=1}^{n} \pi_{i,s,h}(a_i).$$

As a result,

$$
\begin{aligned}
&|r_{i,h}(s, \boldsymbol{\pi}) - r_{i,h}(s, \boldsymbol{\pi}')| \\
&= \left|\sum_{(a_1,\ldots,a_n)\in\mathcal{A}} r_{i,h}(s, \boldsymbol{a}) \prod_{i=1}^{n} \boldsymbol{\pi}_{i,s,h}(a_i) - \sum_{(a_1,\ldots,a_n)\in\mathcal{A}} r_{i,h}(s, \boldsymbol{a}) \prod_{i=1}^{n} \boldsymbol{\pi}'_{i,s,h}(a_i)\right| \\
&= \left|\sum_{(a_1,\ldots,a_n)\in\mathcal{A}} r_{i,h}(s, \boldsymbol{a}) \left(\prod_{i=1}^{n} \boldsymbol{\pi}_{i,s,h}(a_i) - \prod_{i=1}^{n} \boldsymbol{\pi}'_{i,s,h}(a_i)\right)\right| \\
&\leq \sum_{(a_1,\ldots,a_n)\in\mathcal{A}} \left|\prod_{i=1}^{n} \boldsymbol{\pi}_{i,s,h}(a_i) - \prod_{i=1}^{n} \boldsymbol{\pi}'_{i,s,h}(a_i)\right| &\text{(1)} \\
&\leq \sum_{k=1}^{n} \|\boldsymbol{\pi}_{i,s,h} - \boldsymbol{\pi}'_{i,s,h}\|_1 = \|\boldsymbol{\pi}_{s,h} - \boldsymbol{\pi}'_{s,h}\|_1 \\
&\leq \left(\sqrt{\sum_{i=1}^{n} A_i}\right) \|\boldsymbol{\pi}_{s,h} - \boldsymbol{\pi}'_{s,h}\|_2, &\text{(2)}
\end{aligned}
$$

where (1) follows from the fact that $|r_{i,h}(s, \cdot)| \leq 1$ and the triangle inequality. (2) follows from the fact that the total variation distance between two distributions is bounded by the sum of total variation distances between their respective marginal distributions (Hoeffding & Wolfowitz, 1958), and the equivalence between $\ell_1$-norm and $\ell_2$-norm — *i.e.*, $\|\boldsymbol{x}\|_1 \leq \sqrt{m}\|\boldsymbol{x}\|_2$ for $\boldsymbol{x} \in \mathbb{R}^m$).

- the second item is proved using the same line of arguments along with the assumption of additive transitions and the fact that $\left|V_{i,h}^{\boldsymbol{\pi}}(s)\right| \leq H - h$.

□

## C   MORE ON MPGS

Let us complement the previous exposition on MPGs; the main references that we cite are the ones that have provided finite-time computation of approximate NE, (Leonardos et al., 2021; Zhang et al., 2021; Mguni et al., 2021); nevertheless, the same setting is present in other works that considered asymptotic convergence guarantees (Fudenberg & Levine, 1988; Macua et al., 2018). We note some interesting properties of MPGs that further highlight the significance of our results.

**Proposition C.1** ((Zhang et al., 2021)). None of the following conditions imply that an MG is an MPG,

1. There exists a function $\phi : \mathcal{S} \times \mathcal{A}$ in for each state, such that,

$$r_i(s, \boldsymbol{a}) - r_i(s, a_i', \boldsymbol{a}_{-i}) = \phi(s, \boldsymbol{a}) - \phi(s, a_i', \boldsymbol{a}_{-i}), \ \forall s \in \mathcal{S}, \forall \boldsymbol{a}, a_i'.$$

2. There exists a function $\phi : \mathcal{S} \times \mathcal{A}$ such that,

$$r_i(s, a_{-i}', \boldsymbol{a}_{-i}) - r_i(s', a_i'', \boldsymbol{a}_{-i}) = \phi(s, a_{-i}', \boldsymbol{a}_{-i}) - \phi(s', a_i'', \boldsymbol{a}_{-i}), \ \forall s, s' \in \mathcal{S}, \forall \boldsymbol{a}, a_i', a_i''.$$

3. Reward functions are independent of state $s$, such that,

$$r_i(\boldsymbol{a}) - r_i(a_i', \boldsymbol{a}_{-i}) = \phi(\boldsymbol{a}) - \phi(a_i', \boldsymbol{a}_{-i}), \ \forall \boldsymbol{a}, a_i'.$$

The papers referenced (Leonardos et al., 2021; Zhang et al., 2021; Mguni et al., 2021) do not offer an answer regarding the complexity of computing equilibria in these games; assumptions of all three items hold true in our construction in Theorem 3.2 — hence, with no assumption on the transition function, computing approximate nonstationary NEs is PPAD-hard.

# D MISSING PROOFS

## D.1 PROOFS OF SECTION 3.1:HARDNESS

**Theorem D.1.** *Computing a nonstationary Markovian $\epsilon$-approximate NE policy in reward-potential Markov games is* PPAD-*hard.*

## D.2 PROOF OF THEOREM 3.3: NE COMPUTATION IN RPMGS

**Auxiliary lemmata.** There are two key lemmata in the proof of our main theorem; one of them tells us that the game with individual utilities $\left\{ r_{i,h}(s, \cdot) + \sum_{s' \in \mathcal{S}} \mathbb{P}(s'|s, \cdot) V_{i,h+1}(s') \right\}_{i \in [n]}$ is a potential game —w.r.t. policies $\boldsymbol{\pi}_h$ of the corresponding timestep $h$— no matter the (fixed) value vector, $V_{i,h+1}$, of the future states. The second lemma parametrizes the latter games with vectors $V_{i,h+1}$ that correspond to $\delta$-approximate NE for the $\Gamma_{s,h+1}$ subgames; then, it is demonstrated that an $\epsilon$-approximate NE in this game is also a $(\delta + \epsilon)$-approximate NE of the $\Gamma_{s,h}$ subgames.

**Lemma D.1** (Potential game when future values fixed). Fix a timestep $h \in [H]$ and let arbitrary vectors $\{\boldsymbol{v}_i \in \mathbb{R}^{|\mathcal{S}|}\}_{i \in [n]}$. Moreover, for every $s \in \mathcal{S}$ assume game with individual utilities $\{r_{i,h}(s, \cdot) + \sum_{s' \in \mathcal{S}} \mathbb{P}_h(s'|s, \cdot) v_i(s)\}$. Each such game is a potential game.

**Proof.** Indeed, let function $\psi_h(s, \cdot) = \phi_h(s, \cdot) + \sum_{i \in [n]} \sum_{s' \in \mathcal{S}} \omega_{i,s,h} \mathbb{P}_{i,h}(s'|s, \cdot) v_i(s')$. We remind the reader that $\mathbb{P}_h(s'|s, \boldsymbol{\pi}) = \sum_{i \in [n]} \omega_{i,s,h} \mathbb{P}(s'|s, \boldsymbol{\pi}_i)$ due to the additive transitions assumption. It holds for function $\psi_h(s, \cdot)$, that,

$$\psi_h(s, \boldsymbol{\pi}_h) - \psi_h(s, \boldsymbol{\pi}_{i,h}', \boldsymbol{\pi}_{-i,h})$$
$$= \phi_h(s, \boldsymbol{\pi}_h) - \phi_h(s, \boldsymbol{\pi}_{i,h}', \boldsymbol{\pi}_{-i,h}) + \omega_{i,s,h} \sum_{s' \in \mathcal{S}} (\mathbb{P}_{i,h}(s'|s, \boldsymbol{\pi}_{i,h}) v(s') - \mathbb{P}_{i,h}(s'|s, \boldsymbol{\pi}_{i.h}') v(s'))$$
$$= r_{i,h}(s, \boldsymbol{\pi}_h) - r_{i,h}(s, \boldsymbol{\pi}_{i,h}', \boldsymbol{\pi}_{-i,h}) + \omega_{i,s,h} \sum_{s' \in \mathcal{S}} (\mathbb{P}_{i,h}(s'|s, \boldsymbol{\pi}_{i,h}) v(s') - \mathbb{P}_{i,h}(s'|s, \boldsymbol{\pi}_{i.h}') v(s'))$$

The last inequality follows from the reward-potential assumption and completes the proof. $\square$

For brevity, we simplify the notation for the following claim that we need for the promised second lemma.

**Claim D.1** (Approximate best reponses). Let $\hat{\boldsymbol{v}}, \boldsymbol{v}^\dagger \in \mathbb{R}^{\mathcal{S}}$ such that $\|\hat{\boldsymbol{v}} - \boldsymbol{v}^\dagger\|_\infty \le \delta$. Further, let function $r : \mathcal{A} \to \mathbb{R}$ and transition kernel $\boldsymbol{p} : \mathcal{A} \to \Delta(\mathcal{S})$, it holds that,

$$\left| \max_{\boldsymbol{x}' \in \Delta(\mathcal{A})} \left\{ r(\boldsymbol{x}') + \sum_{s' \in \mathcal{S}} p(s'|\boldsymbol{x}') \hat{v}(s') \right\} - \max_{\boldsymbol{x}'' \in \Delta(\mathcal{A})} \left\{ r(\boldsymbol{x}'') + \sum_{s' \in \mathcal{S}} p(s'|\boldsymbol{x}'') v^\dagger(s') \right\} \right| \le \delta.$$

**Proof.** It follows that for every $a \in \mathcal{A}$,

$$r(a) + \sum_{s' \in \mathcal{S}} p(s'|a)\hat{v}(s') - \left( r(a) + \sum_{s' \in \mathcal{S}} p(s'|a)v^\dagger(s') \right) = \sum_{s' \in \mathcal{S}} p(s'|a)\left( \hat{v}(s') - v^\dagger(s') \right) \le \delta.$$

Since the difference,

$$\left| \max_{\boldsymbol{x}' \in \Delta(\mathcal{A})} \left\{ r(\boldsymbol{x}') + \sum_{s' \in \mathcal{S}} p(s'|\boldsymbol{x}')\hat{v}(s') \right\} - \max_{\boldsymbol{x}'' \in \Delta(\mathcal{A})} \left\{ r(\boldsymbol{x}'') + \sum_{s' \in \mathcal{S}} p(s'|\boldsymbol{x}'')v^\dagger(s') \right\} \right|. \quad (3)$$

From linearity, it holds that,

$$\max_{\boldsymbol{x}' \in \Delta(\mathcal{A})} \left\{ r(\boldsymbol{x}') + \sum_{s' \in \mathcal{S}} p(s'|\boldsymbol{x}')\hat{v}(s') \right\} = \max_{a \in \mathcal{A}} \left\{ r(a) + \sum_{s' \in \mathcal{S}} p(s'|a)\hat{v}(s') \right\}$$

and

$$\max_{\boldsymbol{x}'' \in \Delta(\mathcal{A})} \left\{ r(\boldsymbol{x}'') + \sum_{s' \in \mathcal{S}} p(s'|\boldsymbol{x}'')v^\dagger(s') \right\} = \max_{a \in \mathcal{A}} \left\{ r(a) + \sum_{s' \in \mathcal{S}} p(s'|a)v^\dagger(s') \right\}$$

.

The last two displays in combination with (3) which holds for all $a \in \mathcal{A}$ completes the proof of the claim. $\qquad \square$

The last claim proves the following lemma,

**Lemma D.2.** Let $\{\hat{V}_{i,h+1}\}_{i \in [n]}$ be a collection of value vectors that corresponds to a $\delta$-approximate NE, $\{\boldsymbol{\pi}_\tau\}_{\tau \in \{h+1,\ldots,H\}}$, for the subgames $\{\Gamma_{s,h+1}\}_{s \in \mathcal{S}}$. Further, let an $\epsilon$-approximate NE, $\hat{\boldsymbol{\pi}}_h$ of the games with individual utilities $\left\{ r_{i,h}(s,\cdot) + \sum_{s' \in \mathcal{S}} \mathbb{P}_h(s'|s,\cdot)\hat{V}_{i,h+1}(s') \right\}_{i \in [n]}$. Then $\{\boldsymbol{\pi}_\tau\}_{\tau=\{h,\ldots,H\}}$ is a $(\delta + \epsilon)$-approximate NE for subgames $\{\Gamma_{s,h}\}_{s \in \mathcal{S}}$.

**The complexity of implementing the** NE-Oracle. Now, we invoke a theorem that bounds the number of iterations needed to compute an $\epsilon$-approximate NE in a potential game when every player employs the mirror-descent algorithm with a fixed stepsize.

**Theorem D.2** (Theorem B.6 in (Anagnostides et al., 2022)). *Assume a potential game* $\Gamma\left(n, \{\mathcal{A}_i\}_{i \in [n]}, \{u_{i \in [n]}\}\right)$ *with potential function* $\Phi : \prod_{i=1}^n \mathcal{A}_i \to \mathbb{R}$. $\Phi$ *is $L$-Lipschitz continuous. Suppose that each player $i$ employs mirror-descent*

- *with stepsize $\eta = \frac{1}{2L}$,*
- *with regularizer $\mathcal{R}_i(\boldsymbol{x})$, and $\nabla \mathcal{R}_i(\boldsymbol{x})$ $G$-Lipschitz continuous,*
- *and* $\mathrm{Diam}$ *is the maximum diameter of the a player's probability simplex due to their use of regularizer $\mathcal{R}_i$.*

*Further, let* $T = \left\lceil \frac{\eta \Phi_{\max}}{\epsilon^2} \right\rceil + 2$, *then it holds that,* $\exists t^\star \in [T]$, *such that,* $\boldsymbol{x}^{t^\star}$ *is an* $\epsilon\left( \frac{G\mathrm{Diam}}{\eta} + \max_{i \in [n]} \sqrt{|\mathcal{A}_i|} \right)$*-approximate equilibrium.*

**Bounding the total iteration complexity.** Equipped with the latter bound, we are ready to state our bound on the iteration complexity of computing an approximate NE in RPMGs.

**Theorem D.3** (Full version of Theorem 3.3). *Algorithm 1 with* NE-Oracle *implemented using projected gradient descent with stepsize $\eta = \frac{1}{2L}$ for every agent $i \in [n]$, input accuracy $\epsilon/H$ for every $h$, computes an $\epsilon$-approximate nonstationary NE for an RPMG with additive transitions converges with a total number of iterations*

$$\frac{128nH^5|\mathcal{S}|^2 \max_{i \in [n]} |\mathcal{A}_i|^{5/2}}{\epsilon^2}.$$

**Proof.** We remind the reader that the projected gradient descent algorithm is equivalent to mirror descent with $\mathcal{R}_i(\cdot) = \frac{1}{2}\|\cdot\|^2$. Hence, order to achieve accuracy $\epsilon/H$, every projected gradient descent subroutine needs $T = \left\lceil \frac{8L\Phi_{\max}G^2 \text{Diam}^2 \max_{i\in[n]} |\mathcal{A}_i|}{\epsilon^2} \right\rceil + 2$ iterations. In our context, this translates to:

$$T = \left\lceil \frac{128nH^2|\mathcal{S}| \max_{i\in[n]} |\mathcal{A}_i|^{5/2}}{\epsilon^2} \right\rceil + 2.$$

Where we have taken $\text{Diam} = 2\max_{i\in[n]} \sqrt{|\mathcal{A}_i|}$, $G = 1$, $\Phi_{\max} = H$. and we have bounded the Lispchitz-continuity parameter of each $\Gamma_{s,h}$ subgame by $L = 4nH|\mathcal{S}|\max_{i\in n} \sqrt{\mathcal{A}}$ due to Claim B.1. Then, we inductively invoke Lemma D.2 to conclude that after $H$ (backwards) inductive steps, we accumulate an approximation error at most $H\frac{\epsilon}{H} = \epsilon$.

Concluding, we need $|\mathcal{S}|H$ calls to the NE-Oracle with accuracy $\epsilon/H$, raising the total iteration complexity to the stated number.

$\square$

## D.3 PROOFS FOR SECTION 3.2

**Theorem D.4.** *Finite-horizon reward-potential games with additive transitions assert pure Nash equilibria.*

**Proof.** By convention $V_{i,H}(s) = 0, \forall i \in [n], \forall s \in \mathcal{S}$. Further, for $h = H - 1$, the game played in every state $s$ asserts at least one pure Nash equilibrium (Monderer & Shapley, 1996). Then, by Lemma D.1 and Lemma D.2 the claim holds. $\square$

Following using a standard trick we prove the following:

**Corollary D.1.** Infinite-horizon RPMGs with discount parameter $\gamma$, attain a deterministic nonstationary approximate NE that can be computed in time poly $\left(\frac{1}{\epsilon}, \frac{1}{1-\gamma}\right)$.

**Proof.** As proposed in (Daskalakis et al., 2022, Theorem 4.2), the infinite-horizon game can be converted into a finite-horizon one in order to compute nonstationary policies of the initial game. These nonstationary policies of course cannot span the whole horizon of the game; it suffices that they only consider the first $H := \frac{\log(1/\epsilon)}{1-\gamma}$ steps of the game where $\epsilon$ is the desired accuracy of the equilibrium that is sought after.

After truncating the horizon into a finite one, every reward function is scaled according to the initial discounting factor, *i.e.*, $r_{i,h}(s,\cdot) = \gamma^{h-1}r_i(s,\cdot)$, where $r_i(s,\cdot)$ are the reward functions of the infinite-horizon game.

The complexity of computation follows from known results about the computational complexity of pure approximate NE in potential games (Fabrikant et al., 2004) and the use of backwards induction. $\square$

## D.4 PROOFS FOR SECTION 4.1: ARPMGS

First, we prove that although the subgames defined are not adversarial potential games *per se*, the variational inequalities corresponding to their approximate NE coincide with the variational inequalities of a certain adversarial team game.

**Proposition D.1.** Let an ARPMG with additive transitions, $\Gamma(n+1, H, \mathcal{S}, \mathcal{A}, \mathbb{P}, r, \gamma, \boldsymbol{\rho})$, and $\hat{\boldsymbol{V}}_{i,h+1}$ be the value vector for the $\delta$-approximate NE of the subgames $\Gamma_{s,h+1}$. Let the adversarial team normal-form games $\Gamma'_s, \forall s \in \mathcal{S}$, each with $n$ players in the team and one adversary. Define the utility function of the team to be,

$$u(s, \boldsymbol{\pi}) := \phi_h(s, \boldsymbol{\pi}) + \sum_{s'\in\mathcal{S}}\sum_{j\in[n]} \omega_{j,s,h}\,\mathbb{P}_{j,h}(\boldsymbol{\pi}_j)\hat{V}_{j,h+1}(s') - \sum_{s'\in\mathcal{S}} \omega_{\text{adv},s,h}\,\mathbb{P}_{\text{adv},h}(\boldsymbol{\pi}_{\text{adv}})\hat{V}_{\text{adv},h+1}(s').$$

An $\epsilon$-approximate NE of each subgame $\Gamma'_s$ is also an $(\epsilon + \delta)$-approximate NE of the $\Gamma_{s,h}$ subgame.

**Proof.** For brevity, let $\boldsymbol{x}_i \coloneqq \boldsymbol{\pi}_{i,h}, \forall i \in [n]$, with $\boldsymbol{x} \coloneqq (\boldsymbol{x}_1, \ldots, \boldsymbol{x}_n)$, and $\boldsymbol{y} \coloneqq \boldsymbol{\pi}_{\mathrm{adv},h}$. Further, $\mathcal{X} \coloneqq \prod_{i \in [n]} \Delta(\mathcal{A}_i)$ and $\mathcal{Y} \coloneqq \Delta(\mathcal{A}_{n+1})$. Then, we write $u^s(\boldsymbol{\pi}) = u^s(\boldsymbol{x}, \boldsymbol{y})$. An $\epsilon$-approximate NE to the game is computed by solving the following variational inequality problem,

$$\nabla_{\boldsymbol{x}} u(s, \boldsymbol{x}^\star, \boldsymbol{y}^\star)^\top (\boldsymbol{x}^\star - \boldsymbol{x}) \le \epsilon, \forall \boldsymbol{x} \in \mathcal{X} \quad \text{and} \quad \nabla_{\boldsymbol{y}} u(s, \boldsymbol{x}^\star, \boldsymbol{y}^\star)^\top (\boldsymbol{y}^\star - \boldsymbol{y}) \ge -\epsilon, \forall \boldsymbol{y} \in \mathcal{Y}.$$

By computing such a point $(\boldsymbol{x}^\star, \boldsymbol{y}^\star)$, it is also the case that,

$$\nabla_{\boldsymbol{y}} \left( r_{\mathrm{adv},h}(s, \boldsymbol{x}^\star, \boldsymbol{y}^\star) + \sum_{s' \in \mathcal{S}} \mathbb{P}_h(s'|s, \boldsymbol{x}^\star, \boldsymbol{y}^\star) \hat{V}_{\mathrm{adv},h+1}(s') \right) = \nabla_{\boldsymbol{y}} \left( -u(s, \boldsymbol{x}^\star, \boldsymbol{y}^\star) \right)$$

We observe that,

$$\nabla_{\boldsymbol{y}} \left( r_{\mathrm{adv},h}(s, \boldsymbol{x}, \boldsymbol{y}) + \sum_{s' \in \mathcal{S}} \mathbb{P}_h(s'|s, \boldsymbol{x}, \boldsymbol{y}) \hat{V}_{\mathrm{adv},h+1}(s') \right)$$

$$= \nabla_{\boldsymbol{y}} \left( -\phi_{s,h}(\boldsymbol{x}, \boldsymbol{y}) + \sum_{s' \in \mathcal{S}} \mathbb{P}_h(s'|s, \boldsymbol{x}, \boldsymbol{y}) \hat{V}_{\mathrm{adv},h+1}(s') \right)$$

$$= -\nabla_{\boldsymbol{y}} u(s, \boldsymbol{x}, \boldsymbol{y}).$$

By computing such a point $(\boldsymbol{x}^\star, \boldsymbol{y}^\star)$, it is also the case that,

$$\nabla_{\boldsymbol{x}} \left( \phi_h(s, \boldsymbol{x}, \boldsymbol{y}) + \sum_{s' \in \mathcal{S}} \mathbb{P}_h(s'|s, \boldsymbol{x}, \boldsymbol{y}) \hat{V}_{\mathrm{adv},h+1}(s') \right)^\top (\boldsymbol{y}^\star - \boldsymbol{y}) \le \epsilon, \forall \boldsymbol{y} \in \mathcal{Y},$$

$$\nabla_{\boldsymbol{y}} \left( r_{\mathrm{adv},h}(s, \boldsymbol{x}, \boldsymbol{y}) + \sum_{s' \in \mathcal{S}} \mathbb{P}_h(s'|s, \boldsymbol{x}, \boldsymbol{y}) \hat{V}_{\mathrm{adv},h+1}(s') \right)^\top (\boldsymbol{y}^\star - \boldsymbol{y}) \ge -\epsilon, \forall \boldsymbol{y} \in \mathcal{Y}.$$

Concluding, such a strategy $(\boldsymbol{x}^\star, \boldsymbol{y}^\star)$ is also a $(\delta + \epsilon)$-approximate NE for the subgame $\Gamma_{s,h}$. $\quad\square$

This translates to the fact that the template algorithm, Algorithm 1, can be modified in order to compute apprximate NEs for ARPMG using the algorithm proposed in (Anagnostides et al., 2023).

