# OpenReview forum: "Nash Equilibria in Reward-Potential Markov Games: Algorithms, Complexity, and Applications"
_ICLR.cc/2024/Conference — ICLR 2024 Conference Withdrawn Submission_

### Official Review · Reviewer_ku5a · 2023-10-28

**Soundness:** 3 good
**Presentation:** 3 good
**Contribution:** 2 fair
**Rating:** 5
**Confidence:** 4

**Summary:**

This paper considers the reward-potential Markov game instead of the Markov potential game. The paper focuses on the equilibrium computation for such a reward-potential Markov game. The paper then shows that showing $\epsilon$-approximate NE policy in reward potential Markov game is PPAD-hard. Hence, the paper focuses on some variants of the reward-potential Markov game such as additive transition probability, switching control, and single controller.

**Strengths:**

The paper is well-written and the contributions seem to be clear.

The analysis is strong.

**Weaknesses:**

1. The concern is whether the paper has enough contributions to be published at ICLR. The Markov potential game has been relatively well studied. The paper seeks to contribute in the space of reward potential Markov game which only admits potential function for each stage game (which does not mean that it will be MPG as already been shown in the literature). The paper then seeks to find the complexity of computing $\epsilon$ Nash equilibrium. The paper shows that it is PPAD complete. However, the proof is very straightforward from the PPAD-hardness of the $\epsilon$-equilibrium. Given the nature of the paper, it is important to understand the technical novelty.

2. The paper then considers some assumptions such as additive transition probability and then shows that one can find $\epsilon$-equilibrium in O(H^5|S|/\epsilon^2) sample complexity. However, the reviewer is not convinced about the practicality of the algorithm 1. For example, one needs an NE oracle to show the sample complexity. How easily one can find that? Can the algorithm be implemented in a distributed set up.

3. The paper considers that the MDP setup is known which might not be possible in practice.

**Questions:**

1. Does the PPAD-hardness result hold for CCE?

2. Can the authors provide practical examples of additive transition probability assumption holds.

---

### Official Review · Reviewer_qzAJ · 2023-10-31

**Soundness:** 2 fair
**Presentation:** 1 poor
**Contribution:** 1 poor
**Rating:** 3
**Confidence:** 4

**Summary:**

This paper studies the complexity of equilibrium computation in reward-potential Markov games (RPMG). A Markov game is RPMG if there exists a potential function for the reward functions associated with each state. Showing the difficulty of computing (non-stationary) equilibrium even for finite-horizon cases, the authors focused on the additive structure of the underlying transition kernel. They show efficient computation of non-stationary equilibrium in RPMGs with additive transitions for finite-horizon problems and generalize the results to the infinite-horizon cases via the truncation of horizon at the expense of converging near equilibrium rather the exact one.

**Strengths:**

Reward-potential Markov games are of interest.

**Weaknesses:**

- The paper needs proofreading. There are many typos.
- The amount of preliminary information provided in the main draft is too much and too basic. The ones about coarse correlated equilibrium are even redundant since the theoretical results are only about Nash equilibrium. Only the part establishing the notation should stay in the main draft but the other parts should be moved to the Appendix. In that case, the paper will become very short since the contribution (original content) is relatively marginal due to the following reasons:
- - Theorem 3.2 is rather straightforward and intuitive for the arbitrary transition probabilities.
- - The extension through truncation to infinite-horizon problems is also rather straightforward.
- - The additive transition probability structure has not been justified for its practical importance. Once such structures ensure stage games are potential games, computation of the non-stationary equilibrium through backward induction is also very straightforward.
- Proof for Theorem D.1 is missing.

**Questions:**

No clarification is needed.

---

### Official Review · Reviewer_Aoz1 · 2023-11-05

**Soundness:** 3 good
**Presentation:** 2 fair
**Contribution:** 2 fair
**Rating:** 5
**Confidence:** 2

**Summary:**

This paper studies a class of stochastic games named reward potential Markov games (RPMGs) and shows that without additional assumptions on the transitions, the game is PPAD-hard. Then a Nash-VI type algorithm is proposed to efficiently solve RPMGs when the transition is additive. These results are then extended to adversarial RPMGs.

**Strengths:**

The setup of RPMGs is important and the computability results give insights on further designing algorithms for studying RPMGs. The distinctions between RPMGs and potential MGs are also very interesting.

**Weaknesses:**

One of the main contributions of this paper is to extend the existing computability results to the case of RPMG with nonstationary policies. I am not fully convinced why such an extension is non-trivial, or perhaps I am not fully understanding the proof of Theorem 3.2. In a stochastic game, when the number of states is 1 and the length of the time horizon is 1, the problem reduces to an n-player matrix game, which is PPAD-hard. Since RPMGs are a subclass of stochastic games, wouldn't this already verify the hardness of RPMGs? In contrast, I am not sure if it is trivial to extend the existing PPAD-complete results (E.g. Deng 2023) to include nonstationary policies.

On a side note, the paper is very definition-heavy, and the main results are not presented until page 7. I would suggest restructuring the paper, such as moving some common definitions to the appendix.

Minor issues:
1. page 3 typo, "we overload the notation to note$r _{i, h}(s, \boldsymbol{\pi})=\mathbb{E} _{\boldsymbol{a} \sim \boldsymbol{\pi}}\left[r _{i, h}(s, \boldsymbol{a})\right]$ and $\mathbb{P} _h(s, \boldsymbol{\pi})=\mathbb{E} _{\boldsymbol{a} \sim \boldsymbol{\pi}}\left[r _{i, h}(s, \boldsymbol{a})\right]$..."
2. missing reference, I believe (Marden, 2012) is one of the first papers to study the RPMGs.
3. The last line of the introduction, "We then extend our results to the class of adversarial reward-potential MGs (Theorem 3.3)" It should be Theorem 3.4 here.

Marden, J. R. (2012). State based potential games. Automatica, 48(12), 3075-3088.

**Questions:**

1. Algorithm 1 is essentially Nash-VI (value iteration). If I am not mistaken, the complexity for that in two-player zero-sum game is $O(H^3S A_1, A_2 / \epsilon^2)$, where $A_1, A_2$ is the size of the action space for player 1 and 2. How did Theorem 3 remove the dependency on the size of the action space?
2. To my understanding, the hardness (PPAD-hard) result only hinders the implementation matters for the NE oracle (step 4 of algorithm 1). If we do not consider computation issues, do the results for Theorem 3.3 hold for general RPMGs?

---

### Official Review · Reviewer_qkBm · 2023-11-08

**Soundness:** 3 good
**Presentation:** 2 fair
**Contribution:** 2 fair
**Rating:** 5
**Confidence:** 4

**Summary:**

The paper explores the concept of Reward-Potential Markov Games (RPMGs) and their computational complexity. RPMGs are Markov games with rewards exhibiting potential functions in each state. The paper highlights the discrepancy between RPMGs and Markov Potential Games (MPGs) that require state-dependent potential functions for value functions.

The paper shows that computing a stationary Nash equilibrium (NE) in infinite-horizon RPMGs is PPAD-hard, even with constraints on transition functions. Moreover, it establishes that computing a nonstationary NE in finite-horizon RPMGs is PPAD-hard without any assumptions on transition functions. Despite the complexity, the paper presents an algorithm that efficiently computes an ϵ-approximate NE in RPMGs with additive transitions. This algorithm has a runtime that is polynomial in 1/ϵ and the horizon (H) of the game, providing a solution to the "curse of multiagents."  The paper extends its analysis to include an adversarial player seeking to maximize the underlying potential function, introducing the concept of Adversarial Reward-Potential Markov Games.

**Strengths:**

The paper studies an original setup called reward potential Markov games, where the authors show the computational complexity of computing the NE of the game. To some extend, the authors show the solvability of the problem under certain constraints on the dynamics.

The paper have a thorough walkthrough of the definitions of different types games, their relationship, solution concepts, and the ingredients that form a markov game. Such thorough walkthrough will serve a good educational purpose for readers who don't understand this area much.

The proofs in the Appendices are complete, sound and easy to follow. I did not spot any flaws that can undermine the credibility of the paper.

**Weaknesses:**

1. Even though the paper have thorough walkthrough of all the game types, definitions and etc. But the presentation is somewhat lengthy and repetitive. The paper completes the introduction and the definition at page 6, which only leaves 3 pages to show the results. For example, the definition of finite-horizon and infinite-horizon games were separated but their share similar components (policies, rewards) which don't need to be defined twice. The focus on the paper should be Reward-potential Markov games. Let its definition and formulation stands out. I found the discussion of results is rushed, which I will shed more light in the following comments. T

2. Another weakness is the significance of the formulation and results. Algorithm 1 only support RPMGs with additive transitions. There are two major constraints if one needs to apply algorithm: 1. The game has to be  RPGM 2. the transition has to be additive transitions. Throughout the paper, I did not see any discussion about real-world scenarios where RPMGs with additive transitions can successfully model and make an impact.

3. In Theorem 3.1, it is not clear to me from which paper this theorem is from originally, making it hard to check the source theorem.

4. Instead of throwing definitions separately, I'd like to see more discussion about the relationship between RPMGs and PMGs. Provide examples of games that belong to PMGs but not to RPMGs and examples of games that belong to RPMGs but not PMGs.

5. typos throughout the paper. For example, "RPGMs".

**Questions:**

No further questions.

---

### Official Review · Reviewer_kcNT · 2023-11-09

**Soundness:** 3 good
**Presentation:** 3 good
**Contribution:** 3 good
**Rating:** 5
**Confidence:** 3

**Summary:**

This paper is about the complexity of computing a Nash equilibrium in reward-potential Markov games, which are games in which the local reward function in every state obeys a potential function. The authors show that computing Nash equilibrium in this family of games is PPAD-hard, and that it can  be done in polynomial time under a certain assumption (additive transitions).

**Strengths:**

The paper is generally easy to read. The main theorem statements are interesting, and clearly show the distinction in complexity between stationary and non-stationary policies.

**Weaknesses:**

The main gripe concern I have with this paper is that the new setting---additive transitions---seems like almost a trivial generalization of the switching-control setting. Indeed, additive transitions amount to nothing but switching control, but the player in control is chosen at random from the known distribution $\omega_{\cdot, s, h} \in \Delta(n)$. As such I am not sure how significant the main positive result (Thm 3.3) is. For example, was Thm 3.3 previously known for switching-control games? If so, the authors should clarify exactly what (if anything) makes Thm 3.3 not just a trivial generalization of that prior result; if not, the authors should emphasize that their result is also new in the switching-control setting, which would in my opinion strengthen the paper.

In general, the preliminaries section seems to go to great lengths to distinguish between finite- and infinite-horizon games, and between Nash and CCE, but the rest of the paper seems to not really bother too much with these distinctions: the main results (Thms 3.2 and 3.3) both concern Nash equilibrium in games of finite horizon, and the paper doesn't even seem to use CCE at all except in cases where CCE collapses to Nash. Perhaps it is better to frame the paper primarily around Nash in finite-horizon games, and confine the infinite-horizon/CCE case to its own separate section?

It would be nice to have a table illustrating the relevant distinctions in the paper (turn-based vs switching control vs additive; stationary vs non-stationary; potential vs reward-potential vs adversarial reward-potential; finite vs infinite horizon; CCE vs Nash), and the complexity that arises in each case. That is a very large number of different possible combinations, so it would greatly assist a reader if there were such a summary.

There are a large number of simple mistakes/omissions which should certainly be fixed before publication because they are significant enough to change the meanings of some theorem statements. Here is a probably-incomplete list of such issues, in roughly decreasing order of importance. (If any of these are not actually errors, then I have misunderstood something; I would greatly appreciate if the authors would clarify any such misunderstandings if they have happened)

1. Cor 4.1: "is PPAD" should be "is PPAD-hard".
1. Thm 3.3: Should emphasize that the Nash oracle solves a potential game (perhaps even include the statement of Lemma D.1 in the body to emphasize this); otherwise it's not obvious that the algorithm is efficiently implementable. Also, I would not hide a factor of $n \max_i |A_i|^{5/2}$ within a big-O---the dependence on these factors should not be ignored.
1.  Alg 1 should include within its caption something about additive transitions, as that's a necessary condition for the algorithm to work.
1.  Def 2.1, second bullet: coarse correlated equilibrium should be Nash equilibrium
1. Def 2.1, second bullet: $s_1$ should be $\boldsymbol\rho$
1. Def 2.1/2.3, first bullet: What's a "coarse Nash equilibrium"? I'm assuming this is simply a typo and that it's actually defining just a Nash equilibrium
1. Def 2.6: need a negative sign on the adversary's reward
1. pg 7, equation at bottom of page: $0+$ is redundant

**Questions:**

1. In the definition of CCE, is a player allowed to "re-correlate" after a deviation? For example, could a player deviate at a timestep $h$, and then at a later timestep $h' > h$, resume playing according to the correlated profile?
1. What's the point of Proposition 4.1? It seems to be some technical observation, assisting in the proof of Theorem 4.1, that may be better placed in the appendix.